# Endothelial Function and Hypoxic–Hyperoxic Preconditioning in Coronary Surgery with a Cardiopulmonary Bypass: Randomized Clinical Trial

**DOI:** 10.3390/biomedicines11041044

**Published:** 2023-03-28

**Authors:** Irina A. Mandel, Yuriy K. Podoksenov, Sergey L. Mikheev, Irina V. Suhodolo, Yulia S. Svirko, Vladimir M. Shipulin, Anastasia V. Ivanova, Andrey G. Yavorovskiy, Andrey I. Yaroshetskiy

**Affiliations:** 1Department of Anesthesiology and Intensive Care, I.M. Sechenov First Moscow State Medical University (Sechenov University), 119991 Moscow, Russia; 2Department of Anesthesiology and Intensive Care, Federal Scientific and Clinical Center of Specialized Types of Medical Care and Medical Technologies of the Federal Medical and Biological Agency of Russia, 115682 Moscow, Russia; 3Department of Anesthesiology and Intensive Care, Tomsk National Research Medical Center of the Russian Academy of Sciences, Cardiology Research Institute, 634012 Tomsk, Russia; 4Department of Hospital Surgery with a Cardiovascular Surgery Course, Siberian State Medical University, 634050 Tomsk, Russia; 5Department of Cardiovascular Surgery, Tomsk National Research Medical Center of the Russian Academy of Sciences, Cardiology Research Institute, 634012 Tomsk, Russia; 6Department of Internal Medicine Department, Clinical Hospital “MEDSI” in Otradnoe, 123056 Moscow, Russia; 7Department of Morphology and General Pathology, Siberian State Medical University, 634050 Tomsk, Russia; 8Biochemical Laboratory, Tomsk National Research Medical Center of the Russian Academy of Sciences, Cardiology Research Institute, 634012 Tomsk, Russia; 9General Medicine Faculty, I.M. Sechenov First Moscow State Medical University (Sechenov University), 119991 Moscow, Russia; 10Department of Pulmonology, I.M. Sechenov First Moscow State Medical University (Sechenov University), 119991 Moscow, Russia

**Keywords:** oxygen transport, cardioprotection, hypoxic–hyperoxic preconditioning, ischemia-reperfusion injury, cardiopulmonary bypass, endothelial damage markers

## Abstract

A hypoxic–hyperoxic preconditioning (HHP) may be associated with cardioprotection by reducing endothelial damage and a beneficial effect on postoperative outcome in patients undergoing cardiac surgery with cardiopulmonary bypass (CPB). Patients (*n* = 120) were randomly assigned to an HHP and a control group. A safe, inhaled oxygen fraction for the hypoxic preconditioning phase (10–14% oxygen for 10 min) was determined by measuring the anaerobic threshold. At the hyperoxic phase, a 75–80% oxygen fraction was used for 30 min. The cumulative frequency of postoperative complications was 14 (23.3%) in the HHP vs. 23 (41.1%), *p* = 0.041. The nitrate decreased after surgery by up to 20% in the HHP group and up to 38% in the control group. Endothelin-1 and nitric oxide metabolites were stable in HHP but remained low for more than 24 h in the control group. The endothelial damage markers appeared to be predictors of postoperative complications. The HHP with individual parameters based on the anaerobic threshold is a safe procedure, and it can reduce the frequency of postoperative complications. The endothelial damage markers appeared to be predictors of postoperative complications.

## 1. Introduction

Coronary artery bypass graft surgery (CABG) with cardiopulmonary bypass (CPB) is a widely used operation for patients with coronary artery disease. The myocardial ischemic-reperfusion injury (IRI) may reduce the beneficial effect of CABG [1,2]. During cardiopulmonary bypass, systemic inflammation and damage of endothelial cells occur [3,4,5].

Hypoxic–hyperoxic preconditioning (HHP) is an effective nonpharmacological method of increasing the body’s resistance to adverse effects, including IRI [6,7,8,9]. Reduced oxygen delivery during hypoxia increases regional blood flow and functional density of capillaries [10]. Several studies have identified a significant role for hypoxia and hyperoxia signaling in organ protection, including myocardial, respiratory, renal, and intestinal function [11,12,13]. Intermittent hypoxic–hyperoxic procedures increase the resistance of the organism of patients with coronary artery disease [14]. However, the mechanism of implementation of the protective effect of any type of preconditioning remains unclear.

IRI is characterized by endothelial dysfunction, which is associated with insufficient production of nitric oxide (NO), vascular tone dysregulation, and continuous vasoconstriction [15]. An elevated endothelin-1 (ET-1) level is closely associated with the severity of tissue damage and with the incidence of arrhythmias in cases of acute myocardial infarction [16]. On-pump CABG is strongly associated with the ET-1 overproduction [17]. NO plays a key role in the mechanism of oxidative stress, and in the pathophysiology of IRI [18]. The plasma asymmetric dimethylarginine (ADMA) concentration is a marker for endothelial damage associated with the increased risk of cardiovascular complications [19,20]. Recent studies have pointed out that endothelial dysfunction in chronic heart failure patients of the New York Heart Association (NYHA) class II–III have been associated with reduced exercise hyperemia, impaired functional capacity, and increased incidence of hospitalization or death [21].

We hypothesized that a HHP based on changes in inspiratory oxygen fraction (FiO_2_) is associated with a protective effect on myocardial function by reducing the endothelial damage and a beneficial effect on postoperative outcome in patients undergoing CABG with CPB.

## 2. Materials and Methods

### 2.1. Study Design 

A prospective randomized study in parallel groups included consecutively hospitalized patients (*n* = 130) who were scheduled for coronary surgery with CPB (Figure 1).

At the beginning of the operation, patients in the study group (*n* = 60) underwent HHP. In the control group (*n* = 60), the patient’s ventilation parameters were established according to the blood gas analysis to maintain normoxia and normocapnia. The study was approved by the committee on biomedical ethics (June 2015) of Cardiology Research Institute (ClinicalTrials.gov ID: NCT05354648; principal investigator, Irina Mandel; date of registration, 26 April 2022; full protocol available on request). In compliance with the principles of the Declaration of Helsinki, written informed consent was obtained from each patient.

The inclusion criteria: The need for primary coronary artery bypass grafting.

The exclusion criteria: Age over 75 years, urgent surgery, diabetes mellitus, acute exacerbation of any chronic disease one week before surgery, malignancy, aortic regurgitation, intra-aortic balloon pump (IABP) or high doses of catecholamines before surgery, carotid artery stenosis > 70%, paced heart rhythm, NYHA functional class IV, off-pump surgery, known history of obstructive sleep apnea, and a history of an acute coronary event or stroke within 3 months preceding the study. 

### 2.2. The Anaerobic Threshold Measurement

A safe inhaled oxygen fraction for the hypoxic preconditioning phase (10–14% oxygen for 10 min) was determined by measuring the anaerobic threshold. The anaerobic threshold was determined using a “Cardiovit AT-104 PC Ergo-Spiro” (SCHILLER, Baar, Switzerland): After calibration, the spirometer was connected to the patient using a face mask, and the procedure for measuring the oxygen consumption volume (VO2) and the carbon dioxide release volume (VCO2) was performed while breathing room air (21% oxygen). After that, using a hypoxicator, the oxygen content in the inhaled gas mixture was lowered by 2% (i.e., to 19%, followed by 17%, 15%, etc.), and the measurement of VO2 and VCO2 was repeated. The inspired O_2_ lowered incrementally until the anaerobic threshold was attained. The anaerobic threshold was determined at the moment of the intersection of the curves of oxygen consumption and carbon dioxide release. After determining the anaerobic threshold, the percentage of oxygen in the inhaled hypoxic gas mixture was recorded, corresponding to the moment the anaerobic threshold was reached. If the inspired O_2_ at anaerobic threshold was >14%, a 14% O_2_ gas mixture was used during hypoxic preconditioning. If the inspired O_2_ at the anaerobic threshold was between 10 and 14%, the patient breathed 12% O_2_ gas during hypoxic conditioning, and if the anaerobic threshold for O_2_ concentration was <10%, the patient was given 10% O_2_ gas during hypoxic preconditioning.

The second determination of the anaerobic threshold at the beginning of the operation on the background of anesthesia, mechanical ventilation, and myoplegia was carried out according to a specially developed method (please see the Appendix A, describing this technique). The data obtained were compared with the classical method of measuring the anaerobic threshold during ergospirometry and carried out 72 h before the operation.

### 2.3. Anesthesia and Hypoxic–Hyperoxic Preconditioning Procedure

Anesthesia was induced during the patient’s spontaneous breathing with a 6–8% sevoflurane at a fresh gas flow of 8 L/min. Induction of anesthesia with sevoflurane instead of propofol was chosen due to the known anti-preconditioning effect of propofol. All patients were intubated and mechanically ventilated (Primus anesthesia workstation, Dräger Medical, Lübeck, Germany) with a tidal volume based on 6 mL/kg of ideal body weight, a respiratory rate, and a positive end-expiratory pressure (PEEP) of 5–8 cm H_2_O and FiO_2_ to achieve the target values of PaO_2_ 80–120 mmHg and PaCO_2_ 35–45 mm Hg. Anesthesia was maintained by a low-flow technique with sevoflurane 1 MAC, fentanyl infusion of 1–3 μg/kg/h, muscle relaxation with 0.3–0.4 mg/kg/h rocuronium bromide, and propofol infusion of 2–3 μg/kg/h during CPB. There was no ventilation during CPB.

The oxygen saturation by a finger pulse oximeter (SpO_2_) and capnography were monitored continuously. The radial artery was catheterized (20-G Arteriofix; B. Braun Melsungen AG, Germany) for invasive monitoring of blood pressure and obtaining samples. The right internal jugular vein was catheterized (7F Certofix; B. Braun Melsungen AG, Germany). The Swan–Ganz catheter was used to monitor the pulmonary artery pressure and cardiac index. The bladder was catheterized. Arterial and venous blood gases were measured every 10 min after the mechanical ventilation began, during the preconditioning procedure, and until the CPB begins, followed by every 30 min. Acid–base status and plasma ionogram (pH, bicarbonate, lactate, glucose, sodium, potassium, ionized calcium) were measured. The depth of anesthesia was controlled by the bispectral index (at target levels: 40–60).

HHP was administered to anesthetized, catheterized, and mechanically ventilated patients before CPB. HHP includes two phases: A hypoxic phase for 10 min and a hyperoxic phase for 30 min. In a respiratory circuit, a gas mixture with a FiO_2_ 10–14% was created, based on the level of the anaerobic threshold, by supplying air at a rate of 200 mL/min and nitrogen until the required oxygen concentration was reached in the closed circuit. The decrease in FiO_2_ to 10–14% occurred gradually over 2–3 min; this concentration was maintained for 10 min, and then FiO_2_ was increased up to 75–80%, after that the initial parameters of the gas mixture were established until the CPB started.

A gas mixture with FiO_2_ 10–14% provided a decrease in the patient’s PaO_2_ up to 47–52 mm Hg, and a decrease in SpO_2_ up to 85–87%, which is known as a safe and sufficient level for aerobic metabolism [22]. We monitored the cerebral regional saturation (rSO_2_) with a cerebral oximeter (INVOS Somanetics). It was not allowed to fall below the threshold level of 45%.

Patients in the control group were ventilated with the following parameters: Tidal volume based on 6 mL/kg of ideal body weight, a PEEP of 5–8 cm H_2_O and FiO_2_ (21–25%), and the corresponding respiratory rate to maintain PaO_2_ 80–120 mm Hg and PaCO_2_ 35–45 mm Hg.

### 2.4. Perioperative Monitoring

All patients underwent a similar surgical procedure: Coronary artery bypass graft with normothermic CPB. The indication for surgery was atherosclerotic stenosis (>75% of vessel diameter) of three, four, or five coronary arteries. All surgical procedures were performed according to the standard anesthetic protocol using sevoflurane, fentanyl, and muscle relaxants with normothermic CPB (Stockert Instrumente GmbH-DIDECO S.p.A., Munich, Germany—Mirandola, Italy) with a perfusion rate of 2.5 L/min∙m^2^. The CPB connection was carried out following the standard technique according to the aorta-to-right-atrium scheme. We used 3 mg/kg heparin prior to CPB and maintained an activated clotting time >500 s. Mean arterial pressure during CPB was 60–80 mm Hg; central body temperature was 35.5–36.6 °C; hemoglobin level was above 8 g/dL. The prime volume includes 500 mL of 4% modified gelatin solution (Gelofusine; B. Braun Melsungen AG, Melsungen, Germany) and 500 mL of balanced crystalloid solution (Sterofundin Iso; B. Braun Melsungen AG, Germany). We used 2 mg/kg/h tranexamic acid as an antifibrinolytic drug. For cardioprotection, a cold (5C–8C) crystalloid solution (Custodiol HTK-Bretschneider; Dr Franz Kohler Chemie GmbH, Bensheim, Germany) at a dose of 3 mL/kg for 6–8 min via the ascending aorta or coronary artery (in cases of aortic insufficiency) was used. For heparin neutralization, a protamine sulfate at a 1:1 ratio was used.

Vasoactive–inotropic score (VIS) was calculated as follows: (dobutamine mkg/kg/min × 1 + dopamine mkg/kg/min × 1) + 100 × (epinephrine mkg/kg/min + norepinephrine mkg/kg/min). VIS values were calculated immediately as well as 6 h after surgery. Only dopamine (at a dose of 2–7 mkg/kg/min) and norepinephrine (at a dose of 0.1–0.5 mg/kg/min) were used for hemodynamic support (to maintain a mean arterial blood pressure of 60–80 mmHg) in the postperfusion and the postoperative period. Physicians’ titrate vasoactive drugs were blinded to the treatment or control group.

Intraoperative monitoring included continuous electrocardiography, invasive monitoring of blood pressure and central venous pressure, pulse oximetry, capnography, central body temperature (nasopharyngeal sensor), and cardiac index by thermodilution using a Swan–Ganz catheter and Infinity Delta XL monitor (Draeger AG, Lübeck, Germany).

The duration of mechanical ventilation (MV) and length of stay in the intensive care unit (ICU) were recorded.

### 2.5. The Weaning Procedure from Mechanical Ventilation

The criteria for extubation were as follows: PaO_2_/FiO_2_ ratio > 300, SpO_2_ > 92%, a negative inspiratory force of less than −20 cmH_2_O, the absence of exhaustion, agitation, hypertension, and tachycardia, and a withdrawal of sedation, if any.

### 2.6. Oxygen Transport Characteristics

Blood analysis: Arterial oxygen tension (paO_2_), mixed venous oxygen tension (pvO_2_), arterial oxygen saturation (SaO_2_), mixed venous oxygen saturation (SvO_2_), arterial carbon dioxide tension (paCO_2_), mixed venous carbon dioxide tension (pvCO_2_), blood lactate, hemoglobin, creatinine, and glucose levels were measured with Stat Profile CCX (Nova Biomedical, Waltham, MA, USA) initially (immediately after intubation of the trachea) and every 10 min of the preconditioning procedure followed by every 30 min during surgery. 

#### Oxygen Balance Formulas

An arterial oxygen content (CaO_2_) was determined by the formula: (0.0134 × Hb (g/L) × SaO_2_) + 0.0031 × PaO_2_,(1)

A venous oxygen content (CvO_2_) was determined by the formula: (0.0134 × Hb (g/L) × SvO_2_) + 0.0031 × PvO_2_,(2)

The arterio-venous difference (C(a-v)O_2_) in the oxygen content in the blood was determined by the formula: CaO_2_ − CvO_2_,(3)

Cardiac index (CI) was calculated as the cardiac output divided by the body surface area.

The oxygen delivery index (IDO2) was determined by the formula:CaO_2_ × CI, (4)

The oxygen consumption index (IVO2) was determined by the formula: (C(a-v)O_2_) × CI, (5)

An oxygen extraction index was calculated as:IEO_2_ = IVO_2_/IDO_2_ × l00, (6)

We calculated central venous-to-arterial carbon dioxide difference (ΔPCO_2_, the normal ratio is below 6 mm Hg) and central venous-to-arterial carbon dioxide difference/arterial-venous oxygen content difference ratio (ΔPCO_2_/Ca-vO_2_, a cutoff point 1.8) as an additional indicator of perfusion quality and anaerobic metabolism.

### 2.7. Endothelial Damage Markers 

The ET-1 (fmol/mL), ADMA (μmol/L), endogenous nitrite (NO_2_^−^), nitrate (NO_3_^−^), and the total concentration of nitric oxide metabolites (NOx total, μmol/L) were assessed. The analysis of endothelial damage markers included three phases: Baseline, which was 24 h prior to the operation, the end of the operation, and 24 h after the operation. The ET-1 plasma concentration was determined by the Biomedica test system (Vienna, Austria) with a solid-phase enzyme-linked immunosorbent assay and an absorption peak of 450 nm. The limit of sensitivity is 0.02 fmol/mL (0.05 pg/mL). The plasma concentrations of nitric oxide metabolites (nitrites and nitrates) were determined by the R&D Systems Parameter Total NO/Nitrite/Nitrate Kit (Minneapolis, MN, USA) using the enzyme colorimetric method. To avoid differences between the tests, samples of one patient were measured with the same kit. 

### 2.8. Cardiac Troponin T

Cardiac troponin T was determined before and 12 h after surgery (troponin Ths STAT immunoassay by Cobas e 411 analyzer, Roche Diagnostics International Ltd., Mannheim, Germany). The upper reference limit for cTnT, defined as the 99th percentile of healthy participants, is 140 pg/mL. We used the third universal definition of myocardial infarction (MI) that has been defined by MI (type 5 MI) as a 10-times elevation of cardiac troponin during the first 48 h after CABG and as a 7-times elevation for perioperative myocardial injury [23].

### 2.9. Intragastric pH-Metry

Intragastric pH-metry was carried out using a portable acidogastrometer “AGM-03” (NPP “Istok-System”, Fryazino, Russia). The acidogastrometer “AGM-03” is equipped with a three-channel probe, which was inserted in the body of the stomach and its antrum. Intragastric pH (pHg) was measured during surgery (including HHP) and in the early postoperative period. A risk of gut ischemia was predicted with a pHg less than 4.0.

### 2.10. Outcomes

The primary endpoint was the rate of postoperative complications (composite of death, acute myocardial infarction, need for pacemaker, intra-aortic balloon pump, revision of the surgical wound for bleeding, acute kidney injury, delirium, pneumonia, mediastinitis, and stress ulcers of the gastrointestinal tract) during 60 days after surgery.

The secondary endpoints were the length of MV and the vasoactive–inotropic score (VIS) immediately as well as 6 h after surgery, a spontaneous sinus rhythm recovery during 14 days after surgery, cTnT 12 h after surgery, and endothelial damage markers (ET-1, NOx total, NO_2_^−^, NO_3_^−^, ADMA) at 24 h before surgery, at the end of the surgery, and 24 h after surgery. 

### 2.11. Statistical Analysis

Descriptive statistics were used for demographic, laboratory, and clinical parameters for each arm. Continuous and categorical variables were presented as median (inter-quartile range, IQR), number, and percent (%), as appropriate. A comparison of quantitative characteristics between groups was performed using the Mann–Whitney U test. A comparison of the dynamics of quantitative characteristics in each group was performed using Wilcoxon’s test. The χ-square tests (2 × 2) and Fisher’s exact test (if less than ten participants) were performed to assess the significance of the differences between the parameters according to the categorical variables. Differences were considered significant if *p* < 0.05. ROC analysis was performed to assess the prognostic value of factors that were independently associated with postoperative complications. 

The SPSS v28.0.0.0 (IBM, Armonk, NY, USA) was used. The study was reported according to the consolidated standards of reporting trials (CONSORT) guidelines.

Sample size calculation was provided for the study with 130 patients randomized into the two equal groups. Due to the lack of a unified protocol, we developed our protocol of preconditioning, which allows us to simulate the phenomenon of hypoxic–hyperoxic preconditioning in animals and humans [24,25]. The sample size calculation was based on our previous results [24]. The proportion of patients treated with HHP that could meet the criteria for an IRI is estimated at 25% (with 95% confidence interval), and the proportion of control patients could be 42%. Thus, the estimated sample size for the present study was 54 patients in each group. The significance level was 0.05. The power was 80%. The test was 2-sided. Considering a possible 17% dropout rate, sample size was increased to 65 patients in each group. We used a computer-generated permuted block (1:1) randomization sequence (http://sealedenvelope.com, accessed on 1 December 2015). All patients and data analysts were blinded. 

## 3. Results

### 3.1. Preoperative Characteristics of the Patients

The demographic and clinical characteristics of the patients are presented in Table 1. All participants were similar in comorbidities and did not differ between groups.

### 3.2. Intraoperative Characteristics of the Patients

The duration of the operation in the HHP group was median 280 (IQR 246–360) minutes and 280 (250–330) minutes in the control group, *p* = 0.674. The duration of CPB in the HHP group was 112 (93–167) and 110 (96–141) minutes in the control group (*p* = 0.541); the duration of the myocardial ischemia is 80 (69–125) and 80 (71–110) minutes (*p* = 0.860), respectively. The intraoperative and postoperative characteristics are presented in Table 2.

Intragastric pH decreased during CPB, followed by normalization at the end of the surgery in most cases, which was equal in both groups. Additionally, we observed a moderate negative correlation between pHg (at the end of surgery and 6 h after surgery) and CPB and surgery duration. Additionally, low pHg was in conjunction with heart rhythm restoration after CPB through the VF and the need for a pacemaker (Table 3). 

### 3.3. Oxygen Balance Characteristics during Hypoxic–Hyperoxic Preconditioning

The anaerobic threshold was achieved at FiO_2_ 9 (8–11)% in both groups, *p* = 0.863. The anaerobic threshold in 86.7% of patients in HHP group of patients was reached at FiO_2_ < 10% (high threshold), so the hypoxic phase was carried out at 10% oxygen in the inhaled gas mixture; in 10% (*n* = 6) of patients, the anaerobic threshold was moderate, so FiO_2_ 12% was used, and in 3.3% (*n* = 2), the anaerobic threshold was low, and FiO_2_ 14% was used. According to cerebral oximetry, rSO_2_ changed insignificantly. The dynamics of oxygen balance characteristics, rSO_2_, blood glucose, and lactate levels during the HHP are presented in Appendix A. The dynamics of oxygen delivery and consumption during the HHP are presented in Figure 2.

### 3.4. The Dynamics of Endothelial Damage Markers in Blood Plasma 

The dynamics of endothelial damage markers during the perioperative period are presented in Table 4. 

#### The Endothelial Damage Markers as Predictors of Postoperative Complications

We observed statistically significant differences in ET-1 production and a nitric oxide metabolites imbalance in patients with postoperative complications compared with noncomplicated patients (Table 5). 

The ET-1 concentration decreased 24 h after surgery in both groups. The NO_2_ total concentration decreased in both groups, but it was 15% higher in the HHP group at the end of the operation. The nitrate concentration decreased after surgery by up to 20% in the HHP group and up to 38% in the control group. The initial values were restored 24 h after surgery in the HHP group and remained below initial levels in the control group. The endothelial damage markers appeared to be predictors of postoperative complications (Figure 3). 

### 3.5. Heart Rhythm Restoration after Cardiopulmonary Bypass

Before surgery, sinus rhythm was recorded (Holter monitoring) in 81.7% of patients in the HHP group (*n* = 49) and in 89.3% of patients in the control group (*n* = 50), *p* = 0.247. Four patients in each group developed atrial fibrillation (*p* = 0.920). A polytopic ventricular extrasystoles developed in 7 (11.7%) patients in the HHP group, and 2 (3.6%) patients in the control group, *p* = 0.104.

In the postperfusion period, during the conversion from the CPB to physiological circulation, spontaneous sinus rhythm recovery was observed in 34 (56.7%) patients of the HHP group and in 18 patients (32.1%) in the control group, *p* = 0.008 (Figure 4).

### 3.6. The Characteristics of the Postoperative Period

The characteristics of the postoperative period are presented in Table 6. 

### 3.7. Relationship between the ΔPCO_2_, the p(v-a)CO_2_/C(a-v)O_2_ Ratio and Outcome in the HHP Group

The ΔPCO_2_ after 30 min of hyperoxia could serve as a predictor of VF in the postperfusion period; AUC is 0.69 (95% confidence interval (CI) 0.543; 0.833), *p* = 0.019. The cutoff is less than 6.35 µmol/mL, sensitivity—82%, specificity—52% (Figure 5a). The ΔPCO_2_/C(a-v)O_2_ ratio after 10 min of hyperoxia appeared to be a predictor of postoperative complications in the subgroup of patients with chronic heart failure with reduced ejection fraction; AUC is 0.85 (95% confidence interval (CI) 0.682; 1.000), *p* = 0.010. The cutoff is more than 0.90, sensitivity—88%, specificity—64% (Figure 5b).

No harms or unintended effects were observed in either group throughout the trial.

## 4. Discussion

The major findings of our study are a lower complication rate, more frequent spontaneous sinus rhythm recovery, a lower VIS, and a shorter MV in preconditioned patients. 

A comparative analysis of the effectiveness of the HHP in patients undergoing coronary surgery with CPB was conducted. Oxygen transport characteristics were evaluated to control the safety of the procedure. The anaerobic threshold measurement before surgery made it possible to individually select a safely acceptable level of hypoxemia. 

### 4.1. Possible Mechanism of Hypoxic–Hyperoxic Preconditioning

Helmerhorst H. called a “pseudohypoxic” reaction of the body, similar to the true hypoxic reaction, to reduce the oxygen fraction in the respiratory mixture from 70% to 21% [26]. M. Rocco et al. studied the “normobaric oxygen paradox” in which “the relative hypoxia, obtained after a period of hyperoxia, acts as a hypoxic trigger able to significantly increase the erythropoietin or hemoglobin levels” [27]. We provided an analog of such a reaction, during the HHP.

The possible mechanism of hypoxic and hyperoxic combination could be a training of vessels (vasodilation and vasoconstriction in turn).

One of the signs of tissue hypoxia is the dependence of O_2_ consumption on its delivery [28,29,30]. The oxygen delivery system has approximately a threefold reserve of compensatory capabilities, which are used in critical illness to maintain oxygen consumption at a level corresponding to the metabolic needs of the body [18]. However, as soon as compensation in the form of increased O_2_ extraction from the blood reaches its limit, a further decrease in O_2_ delivery leads to a decrease in O_2_ consumption. In these circumstances, oxygen consumption is directly dependent on delivery. This so-called critical level of O_2_ delivery is 330–350 mL/min/m^2^ [30,31]. According to our data, in the hypoxic phase, oxygen delivery did not reach a critical level. In addition, we did not notice any signs of coronary blood flow insufficiency (there were no changes in the ST segment, and cardiac output and hemodynamics were stable). We associate the phenomenon of O_2_ consumption reduction in the process of hyperoxia with the resulting vasoconstriction; similar data were obtained in the works of Mallat J. et al. for vasoconstriction associated with hypocapnia [32]. 

We hypothesized that increased levels of ROS due to the hyperoxic phase of preconditioning boosts and makes longer the effect of the hypoxic phase. Its protective mechanism is implemented along the pseudohypoxic pathway. Exposure to hyperoxia for a short period of time before experimental ischemia induces a mild systemic oxidative stress, which induces a preconditioning-like cardioprotection, reduces the infarction area and arrhithmia [13,22,33,34]. In patients with ST-segment elevation myocardial infarction undergoing coronary artery stenting, infusion of supplemental oxygen directly into the left anterior descending artery (the infarct-related artery) resulted in a reduction in infarct size [35,36,37,38].

### 4.2. Monitoring and Modulation of Tissue Perfusion

Intragastric pH decreased during CPB, followed by normalization at the end of the surgery or later. Additionally, we observed a moderate negative correlation between pHg (immediately and 6 h after surgery) and CPB duration. Therefore, this could be the effect of an altered perfusion of gut mucosa and systemic inflammatory reaction during CPB [39,40].

However, we noted an important shift in the response of the oxygen transport system and microcirculation to hypoxic and hyperoxic exposures. Thus, during the hypoxic phase, there was an increase in VO_2_ with a regular decrease in PvO_2_ and SvO_2_, as well as an increase in tissue perfusion, assessed using ΔPCO_2_ and ΔPCO_2_/C(a-v)O_2_. 

On the contrary, during the hyperoxic phase, opposite changes in PvO_2_ and SvO_2_ were noted after 10 min of hyperoxia, apparently because of the decrease in the tissue perfusion. These changes reached the initial level after the return of normoxia. 

We found that the ΔPCO_2_ after 30 min of hyperoxia can serve as a predictor of VF in the post-CPB period. Thus, this may be a kind of “stress-test” for the adaptive potential of the vessels, where an increase in ΔPCO_2_ could be a sign of transient vasoconstriction and hypoperfusion due to hyperoxia, coinciding with a better heart rhythm recovery after CPB, and vice versa, if ΔPCO_2_ did not change in response to hyperoxia; this could be an indicator of the low adaptive potential of the microcirculation. The ΔPCO_2_ is an indicator of hypoperfusion, either as a result of low cardiac output or vascular pathology. In patients with circulatory shock, it was observed that ventilation at 100% an inhaled oxygen fraction for 5 min increased PvCO_2_ and hence ΔPCO_2_, regardless of changes in the hemodynamic status [41]. This observation may be explained by the lower affinity of hemoglobin for CO_2_ due to increased PvO_2_ (Haldane effect), and it may also reflect impaired capillary blood flow due to the vasoconstrictive effect of hyperoxia [42].

According to Monnet H. and Mallat J., early markers of tissue hypoxia are not only the level of lactate and SvO_2_ but also the ratio ΔPCO_2_/C(a-v)O_2_, at which the value of this marker is more than 1.8 and was associated with the development of anaerobic metabolism and the emergence of the dependence on oxygen consumption and delivery [32,41]. This indicator is analogous to the respiratory quotient (RQ), i.e., the ratio of total CO_2_ production (VCO_2_) to oxygen consumption (VO_2_). ΔPCO_2_ (like VCO_2_) characterizes the relationship between hemodynamics and metabolic processes in the body [42,43,44]. During tissue hypoxia, VCO_2_ increases due to production of CO_2_ from bicarbonate; thus, along with a VO_2_ decrease, RQ and its surrogate (ΔPCO_2_/C(a-v)O_2_) increase [32,41]. According to our data, the elevation of ΔPCO_2_/C(a-v)O_2_ was observed after 10 min of hyperoxia and lasted up to 30 min. These changes might be an indicator of transient vasoconstriction. 

### 4.3. Chronic Heart Failure and Hyperoxia

According to our data, the ΔPCO_2_/C(a-v)O_2_ ratio after 10 min of hyperoxia could serve as an indicator of postoperative complications in patients with chronic heart failure (CHF). If the ΔPCO_2_/C(a-v)O_2_ ratio increased in response to hyperoxia, this could be an indicator of a low adaptive capacity of the vessels. Recently, an opposite vascular reaction to hyperoxia in heart failure by vasodilation due to the inhibition of augmented tonic activity of peripheral chemoreceptors has been described, where patients were exposed to 100% oxygen for 1 min [45]. These differences could be explained by the following: In the mentioned study, the length of exposure was shorter, and 33% of the participants were diabetic patients. Therefore, the vascular reaction could be variable and/or altered not only due to peripheral chemoreceptors but also due to diabetic microangiopathy. Another recent prospective observational study has shown that elevated ΔPCO_2_ and ΔPCO_2_/C(a-v)O_2_ were common phenomena after CPB and cannot be used as reliable indicators to predict the occurrence of organ dysfunction at 48 h after CBP due to the pathophysiological process that occurs after CBP [46].

We suggest that patients suffering from CHF have microvascular abnormalities and endothelial dysfunction, so their vessels react to hyperoxia much quicker (in ten minutes), and even light vasoconstriction will provoke essential hypoperfusion. Thus, the ΔPCO_2_ and the ΔPCO_2_/C(a-v)O_2_ as a response to hyperoxia in a test-like mode may predict complications.

In patients with CHF, an impaired endothelium-dependent relaxation of peripheral arteries was described by several clinical trials. The most likely reason for this is the reduced availability of nitric oxide. In turn, this is associated with a decrease in the activity of the L-arginine–NO synthesis pathway, increased NO degradation by free oxygen radicals, and a decrease in vascular smooth muscles’ responsiveness [21,47].

### 4.4. The ET-1 and NO Metabolites in the Perioperative Period

We observed regular changes in ET-1 and NO metabolites in the perioperative period. A few changes were in conjunction with preconditioning: Less reduction in nitrate concentration at the end of the surgery and stabilization of NO metabolites at 24 h after surgery as compared to the control group. This is contrary to our recent experimental study where we have found that HHP had an infarct-limiting effect, balanced NO metabolites, and reduced ET-1 hyperproduction in the simulation of IRI under the conditions of CPB [25]. Additionally, we observed statistically significant differences in ET-1 production and a NO metabolites imbalance in patients with postoperative complications compared with noncomplicated patients. The mechanisms of vasoconstriction and vasodilation were described by Vignon-Zellweger et al. [48]. In patients with heart failure, ET-1 (vasoconstrictor) is overexpressed, but it may induce the NO (vasodilator) production, prevent apoptosis, and restore cardiac function in surgical stress [48]. Therefore, ET-1 and NO metabolites might be independent predictors of postoperative complications.

### 4.5. Limitations

Several limitations of the current study warrant consideration. This is a single-center study only without mortality. The follow-up period is quite short. A high-risk patient or patients with comorbidities could add a confounding factor to the study. It would be interesting to compare HHP with a remote ischemic preconditioning, the outcomes of which on the heart and the kidney has been investigated extensively in cardiovascular surgery, but clinically some controversy exists regarding its effectiveness.

## 5. Conclusions

In conclusion, HHP before the main stage of coronary surgery, based on the individual selection of parameters using the anaerobic threshold, is a safe procedure; it can reduce the frequency of postoperative complications, the mechanical ventilation time, and the vasoactive–inotropic score. The ΔPCO_2_ after 30 min of hyperoxia could be a predictor of ventricular fibrillation in the post-CPB period. The ΔPCO_2_/C(a-v)O_2_ ratio after 10 min of hyperoxia can serve as a predictor of postoperative complications in patients with chronic heart failure. The endothelial damage markers appeared to be predictors of postoperative complications. The following parameters could predict postoperative complications: the ET-1 24 h before surgery > 0.759 fmol/mL; ET-1 at the end of surgery > 0.710 fmol/mL; NO_2_.endo at the end of surgery < 0.768 µmol/L; NO_2_.total 24 h after surgery > 8.166 µmol/L, and NO3.endo 24 h after surgery > 7.322 µmol/L.

## Figures and Tables

**Figure 1 biomedicines-11-01044-f001:**
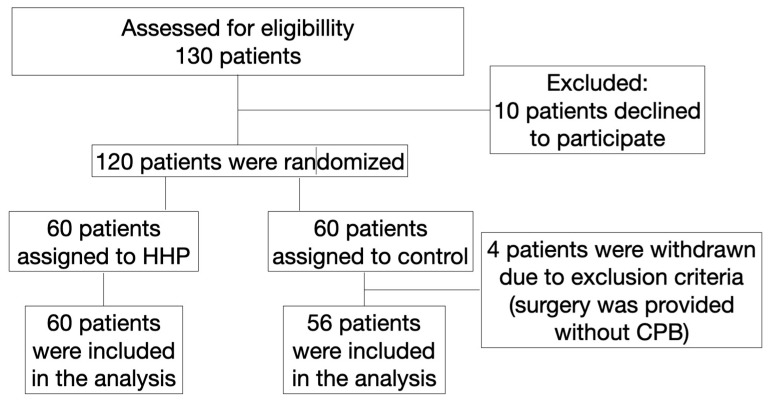
Flowchart of patient enrollment. HHP—hypoxic–hyperoxic preconditioning; CPB—cardiopulmonary bypass.

**Figure 2 biomedicines-11-01044-f002:**
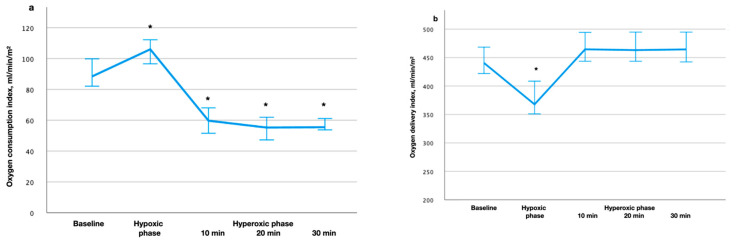
The dynamics of oxygen consumption (**a**) and oxygen delivery (**b**) indices during the hypoxic–hyperoxic preconditioning. *—*p* < 0.05 comparing to the baseline.

**Figure 3 biomedicines-11-01044-f003:**
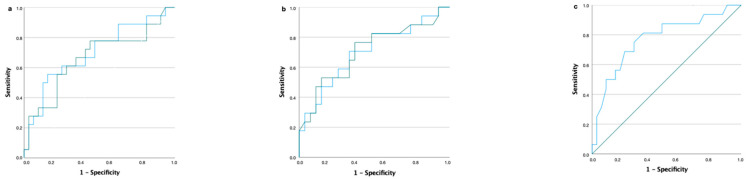
Receiver operating characteristic curves for endothelin-1 and nitric oxide metabolites for the prediction of postoperative complications. (**a**) AUC for ET-1 before surgery (blue line) is 0.70 (95% confidence interval (CI) 0.543; 0.856), *p* = 0.020. The cutoff is 0.759 fmol/mL, sensitivity—78%, specificity—53%; AUC for ET-1 at the end of surgery (green line) is 0.67 (95% CI 0.499; 0.832), *p* = 0.054. The cutoff is 0.710 fmol/mL, sensitivity—78%, specificity—66%. (**b**) AUC for NO_2_.total 24 h after surgery (green line) is 0.71 (95% CI 0.540; 0.872), *p* = 0.023. The cutoff is 8.166 µmol/L, sensitivity—82%, specificity—52%; AUC for NO_3_.endo 24 h after surgery (blue line) is 0.73 (95% CI 0.579; 0.883), *p* = 0.007. The cutoff is 7.322 µmol/L, sensitivity—83%, specificity—52%. (**c**) AUC for NO_2_.endo at the end of surgery is 0.77 (95% CI 0.625; 0.920), *p* = 0.002. The cutoff is 0.768 µmol/L, sensitivity—81%, specificity—66%.

**Figure 4 biomedicines-11-01044-f004:**
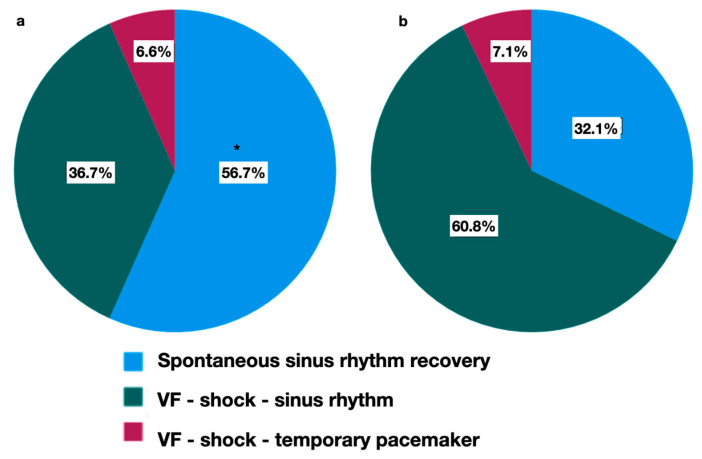
Heart rhythm recovery. The HHP group (**a**) and the control group (**b**) in the postperfusion period. Types of rhythm recovery: (Blue) spontaneous sinus rhythm recovery; (Green) ventricular fibrillation (VF) after aorta declamping, sinus rhythm recovery after defibrillation; (Red) ventricular fibrillation after aorta declamping, atrioventricular blockade after defibrillation, the need for temporary pacing. *—*p* = 0.008 between groups.

**Figure 5 biomedicines-11-01044-f005:**
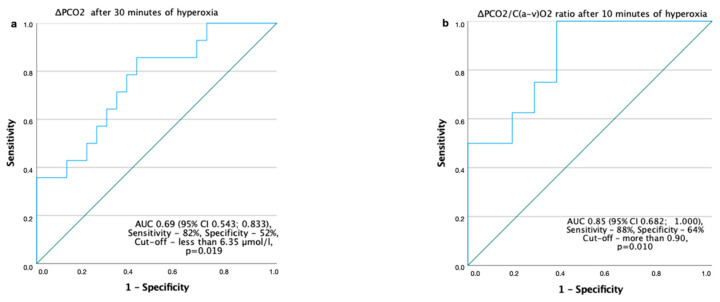
Prediction of complications after surgery by ΔPCO2 and C(a-v)O2 monitoring (ROC curves): (**a**) ΔPCO_2_ after 30 min of hyperoxia for the prediction of ventricular fibrillation after CPB; AUC is 0.69 (95% confidence interval (CI) 0.543; 0.833), *p* = 0.019. The cutoff is less than 6.35 µmol/mL, sensitivity—82%, specificity—52%; (**b**) ΔPCO2/C(a-v)O2 after 10 min of hyperoxia for prediction of postoperative complications in chronic heart failure patients. AUC is 0.85 (95% confidence interval (CI) 0.682; 1.000), *p* = 0.010. The cutoff is more than 0.90, sensitivity—88%, specificity—64%.

**Table 1 biomedicines-11-01044-t001:** Demographic and clinical characteristics of the patients.

Parameter	HHP Group, *n* = 60	Control Group, *n* = 56	*p*-Value
Age, years	59 (55–64)	61 (56; 65)	0.236
Sex (male), *n* (%)	48 (60)	48 (85.7)	0.416
Body mass index, kg/m^2^	27.6 (24.8–31.3)	29.3 (25.5–32)	0.545
EuroSCORE, score (%)	2–6 (1.51–5.89)	2–6 (1.68–4.96)	0.537
Coexisting disease			
Coronary artery disease, *n* (%)	41 (68.3)	36 (64.3)	0.645
Including ejection fraction of left ventricle, %	56 (47.5–62.5)	57.5 (50.8–62.3)	0.826
Coronary artery disease and Heart failure, *n* (%)	19 (31.7)	20 (35.7)	0.645
Including ejection fraction of left ventricle, %	30 (28–36)	33.5 (29.3–37.8)	0.247
Including,			
NYHA I, *n*	1	1	
NYHA II, *n*	15	17	0.850
NYHA III, *n*	3	2	
HFpEF, *n*	0	0	
HFmrEF, *n*	1	1	
HFrEF, *n*	18	19	0.970
Arterial Hypertension, *n* (%)	32 (53.3)	34 (60.7)	0.839
Chronic Obstructive Pulmonary Disease, *n* (%)	16 (26.7)	13 (23.2)	0.668
Pulmonary Hypertension, *n* (%)	9 (15)	12 (21.4)	0.369
Chronic renal disease, *n* (%)	6 (10)	4 (7.1)	0.472
Cerebrovascular disease, *n* (%)	17 (28.3)	15 (26.8)	0.159
Stroke more than 1 year ago, *n* (%)	6 (10)	4 (7.1)	0.480
Gastrointestinal tract pathology, *n* (%)	45 (75)	38 (67.9)	0.104
Smoking, *n* (%)	9 (15)	9 (16.1)	0.750
Stenosis of femoral artheries, %	30 (20–40)	20 (5–33)	0.224
Stenosis of carotid artheries, %	30 (20–55)	30 (13–45)	0.484
Myocardial infarction > 1 year before surgery, *n* (%)	35 (58.3)	39 (69.6)	0.205

NYHA—New York Heart Association; HFrEF—heart failure with reduced ejection fraction ≤40%; HFmrEF—heart failure with mildly reduced ejection fraction 41–49%; HFpEF—heart failure with preserved ejection fraction ≥ 50% (universal definition of heart failure). Values are shown as median (25–75 quartile) or number and percent. *p* values were calculated by Mann–Whitney U test, χ^2^ test, or Fisher’s exact test.

**Table 2 biomedicines-11-01044-t002:** Dynamics of perioperative parameters.

Parameter	HHP Group, *n* = 60	Control Group, *n* = 56	*p*-Value
Mean arterial pressure during CPB, mm Hg	71 (65–75)	60.5 (50–65)	0.395
Mean arterial pressure, lowest during CPB, mm Hg	62 (58–67)	60.5 (50–65)	0.395
Hemoglobin baseline, g/L	130 (121–140)	127 (118–137)	0.171
Hemoglobin, lowest during CPB, g/L	84 (78–92)	85 (77–95)	0.669
Hemoglobin, end of surgery, g/L	103 (92–115)	99 (93–111)	0.326
Creatinine 12 h after surgery, mkmol/L	85 (80.3–94)	85 (74.5–96)	0.701
Troponin T 12 h after surgery, pg/mL	200 (140–290)	221 (129–373)	0.744
pHg baseline	5 (4.2–5.3)	4.5 (4.1–5.2)	0.386
pHg during CPB (lowest)	4 (3.8–4.1)	3.8 (3.7–4.1)	0.174
pHg, end of surgery	4 (3.7–4.3)	4 (3.7–4.3)	0.987
pHg, 6 h after surgery	4.1 (4.0–4.7)	4.1 (4.0–4.7)	0.934
Intraoperative blood loss, mL	350 (300–450)	350 (300–450)	0.934

CPB—cardiopulmonary bypass; pHg—intragastric pH. Values are shown as median and [25; 75 quartile]. Values were calculated using the Mann–Whitney test.

**Table 3 biomedicines-11-01044-t003:** Correlation between the intragastric pH level and CPB and surgery duration and type of postperfusion rhythm recovery.

	Po Spearmen’s	95% Confidence Interval	*p*-Value
pHg, end of surgery—length of surgery	−0.409	−0.612; −0.156	0.002
pHg, 6 h after surgery—legnth of surgery	−0.366	−0.579; −0.106	0.006
pHg, end of surgery—length of CPB	−0.267	−0.501; 0.004	0.047
pHg during CBP—Heart rhythm restoration after CPB (spontaneous sinus rhythm or VF and need for pacemaker)	−0.351	−0.567; −0.089	0.008

pHg—intragastric pH; CPB—cardiopulmonary bypass; VF—ventricular fibrillation. Spearmen’s correlation test.

**Table 4 biomedicines-11-01044-t004:** Characteristics of endothelin-1, nitric oxide metabolites, and ADMA concentrations in the perioperative period.

Parameter	HHP, *n* = 60	Control, *n* = 56	*p*-Value #
Changes in ET-1 concentration			
ET-1, before surgery, fmol/mL	0.785 (0.532–1.465)	0.879 (0.691–1.888)	0.275
ET-1, end of surgery, fmol/mL	0.804 (0.469–1.415)	0.876 (0.521–1.511) *p* = 0.028 *	0.194
ET-1, 24 h after surgery, fmol/mL	0.549 (0.445–0.969) *p* = 0.019 *	0.626 (0.316–1.283) *p* = 0.001 *	0.495
Changes in NO_2_ total concentration			
NO_2_ total, before surgery, µmol/L	10.959 (8.715–16.785)	11.788 (9.393–13.741)	0.815
NO_2_ total, end of surgery, µmol/L	8.355 (6.815–11.527) *p* = 0.001 *	7.244 (5.846–9.530) *p* = 0.001 *	0.190
NO_2_ total, 24 h after surgery, µmol/L	9.165 (6.701–11.609) *p* = 0.005 *	10.146 (6.450–13.053) *p* = 0.044 *	0.511
Changes in nitrite concentration			
NO_2_ endo, before surgery, µmol/L	1.004 (0.666–1.574)	0.878 (0.631–1.574)	0.848
NO_2_ endo, end of surgery, µmol/L	0.760 (0.474–1.048) *p* = 0.039 *	0.809 (0.525–0.967)	0.992
NO_2_ endo, 24 h after surgery, µmol/L	0.913 (0.649–1.118)	0.874 (0.641–1.941)	0.729
Changes in nitrate concentration			
NO_3_ endo, before surgery, µmol/L	8.649 (6.063–14.071)	10.153 (8.235–12.254)	0.294
NO_3_ endo, end of surgery, µmol/L	6.934 (5.831–9.559) *p* = 0.001 *	6.330 (5.478–8.324) *p* = 0.002 *	0.476
NO_3_ endo, 24 h after surgery, µmol/L	8.206 (5.861; 10.215)	9.290 (6.168–11.182) *p* = 0.025 *	0.428
Changes in ADMA concentration			
ADMA, before surgery, µmol/L	0.744 (0.584–0.951)	0.606 (0.536–0.800)	0.318
ADMA, end of surgery, µmol/L	0.776 (0.729–0.809)	0.697 (0.551–0.820)	0.036
ADMA, 24 h after surgery, µmol/L	0.848 (0.757–0.938)	0.794 (0.626–0.999)	0.084

HHP—hypoxic–hyperoxic preconditioning; control (without preconditioning); ET-1—endothelin 1; NO_2_.endo—endogenous nitrite (NO_2_); NO_3_.endo—endogenous nitrate (NO_3_); ADMA—asymmetric dimethylarginine. The results are given as median (25–75 quartile). *—*p*-value of intragroup analysis compared to the initial data (before surgery), the Wilcoxon test; #—*p*-value of between group comparison, the Mann–Whitney test.

**Table 5 biomedicines-11-01044-t005:** Characteristics of endothelin-1, nitric oxide metabolites, and ADMA concentrations in patients with or without postoperative complications.

Parameter	Complicated Patients, *n* = 43	Noncomplicated Patients, *n* = 73	*p*-Value
ET-1, before surgery, fmol/mL	1.506 (0.702–4.019)	0.759 (0.577–1.120)	0.049
ET-1, end of surgery, fmol/mL	1.217 (0.715–4.075)	0.665 (0.510–1.168)	0.053
NO_2_ endo, end of surgery, µmol/L	0.479 (0.410–0.749)	0.852 (0.651–1.093)	0.002
ADMA, end of surgery, µmol/L	0.715 (0.531–0.772)	0.793 (0.735–0.862)	0.045
NO_2_ total, 24 h after surgery, µmol/L	12.038 (8.754–14.041)	8.093 (5.695–10.853)	0.023
NO_3_ endo, 24 h after surgery, µmol/L	10.504 (7.461–12.412)	7.317 (5.550–9.825)	0.007

ET-1—endothelin 1; NO_2_.endo—endogenous nitrite (NO_2_); NO_3_.endo—endogenous nitrate (NO_3_); ADMA—asymmetric dimethylarginine. The results are given as median [Q25; Q75]. *p*-value of between group comparison, the Mann–Whitney test.

**Table 6 biomedicines-11-01044-t006:** Characteristics of the postoperative period.

Parameter	HHP Group, *n* = 60	Control Group, *n* = 56	*p*-Value
Number of complications (cumulative), *n* (%)	14 (23.3)	23 (41.1)	0.041
VIS after CPB, score	6 (5–70)	9.5 (6.3–15)	0.001
VIS 6 h, score	3 (0–6)	6 (5–11)	0.001
VIS 24 h, score	0 (0–3)	0 (0–3)	0.929
Need for pacing (temporary), *n* (%)	3 (5)	6 (10.7)	0.251
Length of mechanical ventilation, h	10 (8–22)	17 (11.3–24)	0.015
Need for intra-aortic balloon pump, *n* (%)	2 (3.3)	2 (3.6)	0.944
Revision for bleeding, *n* (%)	0	1 (1.8)	
Myocardial infarction, *n* (%)	0	0	
Gastrointestinal dysfunction (stress ulcer bleeding, gastrostasis, intestinal hypomotility), *n* (%)	3 (5.0)	6 (10.7)	0.251
Respiratory complications (ARDS, pneumonia), *n* (%)	1 (1.7)	4 (7.1)	0.148
Pulmonary hypertension, *n* (%)	1 (1.7)	0	0.332
Acute renal failure, *n* (%)	2 (3.3)	1 (1.8)	0.600
Delirium, *n* (%)	1 (1.7)	2 (3.6)	0.519
Mediastinitis, *n* (%)	1 (1.7)	1 (1.8)	0.961
Length of stay in ICU, days	1 (1–2)	1 (1–2)	0.507
Length of hospital stay, days	12 (11–14)	12 (11.3–14)	0.161

ARDS—acute respiratory distress syndrome. VIS—vasoactive–inotropic score, CPB—cardiopulmonary bypass. Numerical data are expressed as median (IQR); categorical data are shown as the number of cases (percentage). *p* was pointed on between group comparison, the Mann–Whitney test, and Xi square test or Fisher’s exact test, as appropriate.

## Data Availability

The datasets analyzed during the current study are available from the corresponding author on reasonable request.

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
