# Peer review of "Endothelial Function and Hypoxic–Hyperoxic Preconditioning in Coronary Surgery with a Cardiopulmonary Bypass: Randomized Clinical Trial"

_biomedicines, 2023, doi:10.3390/biomedicines11041044_

Round 1

Reviewer 1 Report

This study tested the hypothesis that aypoxic-hyperoxic preconditioning (HHP) is associated with a protective effect on myocardial function by reducing endothelial damage and a beneficial effect on postoperative outcome in patients undergoing cardiac surgery with cardiopulmonary bypass (CPB). The manuscript is intriguing and well written. The major strengths are the power sample calculation, the quest for a standardized preconditioning and accurate description of methods as well as of the results in the context of published evidence. The major limitation is the lack of procedural data to prove similar extent and invasiveness of surgery and a more clear clinical bottom line as far as study implications are concerned. All in all it needs rework and rereview.

Author Response

Response to Reviewer 1 Comments

This study tested the hypothesis that hypoxic-hyperoxic preconditioning (HHP) is associated with a protective effect on myocardial function by reducing endothelial damage and a beneficial effect on postoperative outcome in patients undergoing cardiac surgery with cardiopulmonary bypass (CPB). The manuscript is intriguing and well written. The major strengths are the power sample calculation, the quest for a standardized preconditioning and accurate description of methods as well as of the results in the context of published evidence. The major limitation is the lack of procedural data to prove similar extent and invasiveness of surgery and a more clear clinical bottom line as far as study implications are concerned. All in all it needs rework and rereview. 

Response 1: Thank you for your valuable comments and support of our manuscript! We introduced the corresponding changes to the methods and resuts sections. 

Reviewer 2 Report

This paper studied the Endothelial function and Hypoxic-hyperoxic preconditioning coronary surgery with a cardiopulmonary bypass. The research is meanful for the safe prediction of cornary surgery. But the paper need to be carful revised before public. My suggestions are as follows:

1. A lot of formatting problems, for example, upper and lower scripts in chemical formulas.

2. The letters on the drawing is too small. Picture quality needs to be improved

3. The Discussion section can be divided into several subsections with illustrations.

4. The background introduction in the abstract needs to be modified to make the sentence more attractive to the reader.

5. Modify the statement expression of the whole manuscript to make the statement more concise and logical.

Author Response

Response to Reviewer 2 Comments

This paper studied the Endothelial function and Hypoxic-hyperoxic preconditioning coronary surgery with a cardiopulmonary bypass. The research is meanful for the safe prediction of cornary surgery. But the paper need to be carful revised before public. My suggestions are as follows:

Point 1: A lot of formatting problems, for example, upper and lower scripts in chemical formulas.

Response 1: Thank you for your valuable comments and support of our manuscript! We introduced the corresponding changes to the manuscript.

Point 2: The letters on the drawing is too small. Picture quality needs to be improved

Response 2: We have redrawn the pictures

Point 3: The Discussion section can be divided into several subsections with illustrations.

Response 3: We introduced the subsections to the discussion section. We placed the illustrations into the results section.

Point 4: The background introduction in the abstract needs to be modified to make the sentence more attractive to the reader.

Response 4: We introduced the corresponding changes to the abstract and introduction.

Point 5: Modify the statement expression of the whole manuscript to make the statement more concise and logical.

Response 5: We introduced the corresponding changes to the manuscript.

Reviewer 3 Report

The paper Biomedicines-2249887 describes a small, randomized clinical trial on hypoxic-hyperoxic preconditioning (HHP) in patients undergoing elective coronary artery bypass grafting (CABG) with the use of cardiopulmonary bypass. The clinical trial follows animal experiments in rabbits, performed by the same group (Int. J. Mol. Sci. 2020; 21: 5336; doi:10.3390/ijms21155336) and builds on the idea of ischemic preconditioning of the myocardium, launched in 1986 (Circulation 1986;74:1124–1136) which has generated a lot of interest but essentially no clinical application. In this trial, HHP consisted of 10 minutes of breathing a gas mixture containing 10-14% O2 and 30 min of breathing 75-80% O2 in anesthetized, intubated patients immediately prior to CABG. The study has been registered as ClinicalTrials.gov ID: NCT05354648.

The authors report that the cumulative frequency of postoperative complications in the HHP group was lower than in the control group (14 (23.3%) vs. 23 (41.1%), p=0.041). Plasma nitrate, a marker of NO-synthase activity, decreased after surgery by up to 20% in the HHP group and up to 38% in the control group, but differences between groups were not significant. The only significant differences between the HHP group and the control group were lower values of the vasoactive-inotropic score immediately after surgery and after 6 hours, shortened time of mechanical ventilation, and a greater percentage of spontaneous sinus rhythm recovery after surgery.

My concern is that the outcomes described in the paper are not the same as prespecified in the clinical trial registration. The authors should discuss the reasons for these differences and explicitly state the lack of differences in primary end-points (even by the broadened definition) in the Abstract and in the Conclusions.

p.s.

In the paper  Biomedicines-2249887 (lines 231-238), the authors give the following definition of the outcomes:

The primary endpoint was the rate of postoperative complications (composite of death, acute myocardial infarction, need for pacemaker, intra-aortic balloon pump, revision  of the surgical wound for bleeding, acute kidney injury, delirium, pneumonia, mediastinitis, stress ulcers of the gastrointestinal tract) during 60 days after surgery.

The secondary endpoints were the length of mechanical ventilation and vasoactive inotropic score (VIS)  immediately as well as 6 h after surgery, spontaneous sinus rhythm recovery during 14days after surgery, cTnT 12 h after surgery, and endothelial damage markers (ET-1, NOx, total, NO2−, NO3−, ADMA) at 24 h before surgery, end of the surgery, 24 h after surgery.

The original, prespecified outcomes of the registered clinical trial NCT05354648 (https://clinicaltrials.gov/ct2/show/NCT04833283?intr=Hypoxic+hyperoxic+preconditioning&cntry=RU&draw=2&rank=1):

Primary Outcome Measures :

  1. Death from a cardiac cause [ Time Frame: The first 30 days from the date of discharge ] Death from cardiac causes during the current hospitalization or 30 days after discharge

  1. Myocardial infarction [ Time Frame: through study completion, an average of 1 month after discharge ] Myocardial infarction after current surgery and one month after discharge

  1. Stroke [ Time Frame: through study completion, an average of 1 month after discharge ] Stroke in postoperative period until the date of the discharge

Secondary Outcome Measures :

  1. Electrocardiogram deterioration caused by myocardial ischemia; [ Time Frame: through study completion, an average of 1 month after discharge ] Electrocardiogram deterioration caused by myocardial ischemia (ST segment elevation or depression, Q wave, negative T wave ) during hospitalization or 30 days after discharge

  1. Angina pectoris - typical chest pain III - IV functional class (Canadian Cardiovascular Society classification) [ Time Frame: through study completion, an average of 1 month after discharge ] Having an angina pectoris during hospitalization or 30 days after discharge

  1. Documented episodes of atrial fibrillation/ atrial flutter. [ Time Frame: through study completion, an average of 1 month after discharge ] Documented episodes of atrial fibrillation or atrial flutter during hospitalization or 30 days after discharge

  1. Ventricular proarrhythmia requiring additional therapy [ Time Frame: through study completion, an average of 1 month after discharge ] Detection of ventricular proarrhythmia during hospitalization or 30 days after discharge

  1. Atrioventricular block 2-3 degrees [ Time Frame: through study completion, an average of 1 month after discharge ] Atrioventricular block 2-3 degree during hospitalization or 30 days after discharge

  1. Episodes of hypotension requiring additional therapy [ Time Frame: through study completion, an average of 1 month after discharge ] Episodes of hypotension (systolic blood pressure of less than 90 mmHg/ diastolic of less than 60 mmHg) requiring additional therapy during hospitalization or 30 days after discharge

  1. Length of stay in the intensive care unit [ Time Frame: through study completion, an average of 1month ]

Author Response

Response to Reviewer 3 Comments

The paper Biomedicines-2249887 describes a small, randomized clinical trial on hypoxic- hyperoxic preconditioning (HHP) in patients undergoing elective coronary artery bypass grafting (CABG) with the use of cardiopulmonary bypass. The clinical trial follows animal experiments in rabbits, performed by the same group (Int. J. Mol. Sci. 2020; 21: 5336; doi:10.3390/ijms21155336) and builds on the idea of ischemic preconditioning of the myocardium, launched in 1986 (Circulation 1986;74:1124–1136) which has generated a lot of interest but essentially no clinical application. In this trial, HHP consisted of 10 minutes of breathing a gas mixture containing 10-14% O2 and 30 min of breathing 75-80% O2 in anesthetized, intubated patients immediately prior to CABG. The study has been

registered as ClinicalTrials.gov ID: NCT05354648.

The authors report that the cumulative frequency of postoperative complications in the HHP group was lower than in the control group (14 (23.3%) vs. 23 (41.1%), p=0.041). Plasma nitrate, a marker of NO-synthase activity, decreased after surgery by up to 20% in the HHP group and up to 38% in the control group, but differences between groups were not significant. The only significant differences between the HHP group and the control group were lower values of the vasoactive-inotropic score immediately after surgery and after 6 hours, shortened time of mechanical ventilation, and a greater

percentage of spontaneous sinus rhythm recovery after surgery.

- My concern is that the outcomes described in the paper are not the same as prespecified in the clinical trial registration. The authors should discuss the reasons for these differences and explicitly state the lack of differences in primary end-points (even by the

broadened definition) in the Abstract and in the Conclusions.

p.s.

In the paper Biomedicines-2249887 (lines 231-238), the authors give the following

definition of the outcomes:

The primary endpoint was the rate of postoperative complications (composite of death, acute myocardial infarction, need for pacemaker, intra-aortic balloon pump, revision of the surgical wound for bleeding, acute kidney injury, delirium, pneumonia, mediastinitis, stress ulcers of the gastrointestinal tract) during 60 days after surgery.The secondary endpoints were the length of mechanical ventilation and vasoactive inotropic score (VIS)

immediately as well as 6 h after surgery, spontaneous sinus rhythm recovery during 14days after surgery, cTnT 12 h after surgery, and endothelial damage markers (ET-1, NOx, total, NO2−, NO3−, ADMA) at 24 h before surgery, end of the surgery, 24 h after surgery. The original, prespecified outcomes of the registered clinical trial NCT05354648 (https://clinicaltrials.gov/ct2/show/NCT04833283?

intr=Hypoxic+hyperoxic+preconditioning&cntry=RU&draw=2&rank=1):

Response 1: Thank you for your valuable comments and support of our manuscript! Unfortunately, the link is not related to our study. The relevant link is as follows https://clinicaltrials.gov/ct2/show/NCT05354648? cond=Hypoxic+hyperoxic+preconditioning&draw=2&rank=1

Please, see the files below.

Round 2

Reviewer 1 Report

The revised manuscript is now worth of publication 

Author Response

Thank you for support of our manuscript! We highly appreciate your help!

Reviewer 2 Report

The paper can be accept after finish the improvement of figure quality. The letters on the figures still too small.

Author Response

Thank you for support of our manuscript! We highly appreciate your help! We improved the figures and attached them separately, we added figures descriptions also.

Reviewer 3 Report

The authors avoided my question. 

The two links cited in their response  refer to nearly identical study designs with different  end-points. The authors seem to have registered their study again after completion. This deserves an explanation.

Author Response

Response to Reviewer 3 Comments 

The authors avoided my question. 

The two links cited in their response  refer to nearly identical study designs with different  end-points. The authors seem to have registered their study again after completion. This deserves an explanation.

Response: Thank you for support of our manuscript! We highly appreciate your help!

We apologize for the inconvenience due to the registration of the study.

Our study (NCT05354648 «Effects of Hypoxic-hyperoxic Preconditioning in Cardio-surgical Patients») was conducted in 2015-2016, patients underwent coronary surgery in Cardiology Research Institute. Recently, I changed my place of work to Sechenov university. The study was registered retrospectively in 2022. The preconditioning procedure was conducted in operating theatre during surgery. Oxygen fractions of 10-14% and 75-80% were used. 

In study NCT04833283 «The Effects of Intermittent Hypoxic-hyperoxic Preconditioning for Patients Undergoing Cardiopulmonary Bypass» oxygen fractions of 12% and 35% were used. The preconditioning procedure was provided before surgery: 4 trainings daily. Patients underwent aortic or mitral valves replacement.

These are two different studies. Nothing in common, except for similar titles and affiliations of researchers.

Round 3

Reviewer 3 Report

The authors have answered my concerns.